# Psychotic Episode and Schizophrenia in Slovakian Narcolepsy Database

**DOI:** 10.3390/brainsci13010043

**Published:** 2022-12-24

**Authors:** Eva Feketeová, Jozef Dragašek, Katarína Klobučníková, Peter Ďurdík, Simona Čarnakovič, Miriam Slavkovská, Martina Chylová

**Affiliations:** 1Department of Neurology, Faculty of Medicine, University of P.J. Šafárik and University Hospital of L. Pasteur, Trieda SNP 1, 04011 Košice, Slovakia; 21st Department of Psychiatry, Faculty of Medicine, University of P.J. Šafárik and University Hospital of L. Pasteur, Trieda SNP 1, 04011 Košice, Slovakia; 31st Department of Neurology, Faculty of Medicine, Comenius University and University Hospital in Bratislava, Mickiewiczova 13, 81369 Bratislava, Slovakia; 4Clinic of Children and Adolescents, Jessenius Faculty of Medicine, Comenius University in Bratislava and University Hospital in Martin, Kollarova 2, 03659 Martin, Slovakia

**Keywords:** psychotic episode, schizophrenia, narcolepsy, hypocretin deficiency

## Abstract

Narcolepsy type 1 (NT1), a central disorder of hypersomnolence, is associated with mood, anxiety or hyperactivity mental disorders. Association with psychotic episode or schizophrenia is rare and could be the source of diagnostic and therapeutic difficulties. Their frequency in the national narcolepsy database has not been systematically studied. The aim of the presented study was to calculate the frequency of NT1 patients diagnosed with psychosis and/or schizophrenia, to identify clinical characteristics of these cases, and to look for narcoleptic and psychotic symptoms during re-evaluation years later. We identified three (4%) cases diagnosed with a psychotic episode in the course of NT1. They were diagnosed with NT1 by age ≤18 years. In the re-evaluation (mean follow-up 9.8 years), we identified one case with a dual diagnosis of NT1 and schizophrenia; two cases were diagnosed with a solitary psychotic episode in the course of NT1. NT1 patients diagnosed in the age ≤18 years are at higher risk of psychotic episode, and this may be related to higher vulnerability during the ongoing neurodevelopmental period. Comorbid schizophrenia with NT1 in the Slovakian Narcolepsy Database was within the prevalence expected in the general population. The solitary psychotic episode in the course of NT1 did not reduce the possibility of subsequent symptomatic treatment afterwards.

## 1. Introduction

Narcolepsy is a rare sleep–wake disorder that belongs to central hypersomnias. NT1 is a subtype with the specific phenotype of excessive daytime sleepiness (EDS) lasting at least 3 months and cataplexy with/without hypnagogic/hypnopompic hallucinations, sleep paralysis, vivid dreams, and disturbed night sleep [1]. The diagnosis is confirmed by night and daytime polysomnographic studies (PSG) searching for REM sleep abnormalities: Sleep Onset REM (SOREM) in night sleep and mean sleep latency and number of SOREMs in the Multiple Sleep Latency Test (MSLT). An alternative diagnostic tool is an examination of Hypocretin- 1 (Hcrt1) in cerebrospinal fluid [2]. Low or undetectable Hcrt1 is tightly connected with the phenotype of narcolepsy with cataplexy, which is very important in cases when PSG cannot be done under the standard conditions or shows borderline results [3].

Hcrt1 deficiency is hypothesised to be an autoimmune disease, even if it does not fully meet the criteria for being classified as a genuine autoimmune disease [4]. The strongest genetic predisposition focusing on autoimmune origin is formed by the specific haplotype HLA DQB1*0602 [5], which is present in nearly 100% of cases of Hcrt1 deficient narcolepsy. Despite major advances in understanding of the neurobiological basis of NT1, therapy remains nowadays only symptomatic [6]. If some biomarkers could specifically target, for example, the treatment of obesity based on distinct metabolic pathways in NT1 patients, then this has to be confirmed in the future [7]. Hcrt/orexin-based therapy provides another challenging therapeutic option, especially nonpeptide orexin receptor agonists that are currently under development [7]. The role of immune-based therapies administered as early as possible after disease onset could theoretically slow down the destruction of orexin neurons in some selected patients [7].

Patients with NT1 also present overlapping symptoms with psychiatric disorders, such as anxiety and depressed mood, and even hallucinations, which could lead to misdiagnosis in the initial assessment. Despite the different characteristics in the type of auditory hallucinations and the higher frequency of visual hallucinations in NT1 [8], determining their origin may not be easy. Other REM sleep abnormalities such as vivid dreams, nightmares and REM sleep behaviour disorder observed in 20–70% of NT1 patients [9] could be a source of diagnostic challenges as well. While robust evidence suggests a connection between the hypocretin/orexin system and major depressive disorder, findings of hypocretin involvement in schizophrenia are inconsistent [10,11]. Rare case reports in the literature and our clinical experience of NT1 accompanied by a psychotic disorder gave us the idea to look up their frequency in the Slovak Narcolepsy Database. This study aimed to identify the prevalence of psychosis in patients with NT1 and to examine affected individuals many years after diagnosis to determine whether psychotic symptoms were a single-episode manifestation of NT1 or were part of well-established schizophrenia.

## 2. Materials and Methods

The Slovak Narcolepsy Database combines data from three Sleep centres in Slovakia. It includes patients with a confirmed diagnosis of NT1 according to ICSD2/3 [2,12]. All patients diagnosed with narcolepsy in the sleep labs are referred for a second opinion to the sleep laboratory and centre for rare hypersomnias at the University Hospital of L. Pasteur in Kosice. Patients who signed an informed consent (at the moment 100%) undergo an extended interview focused on the details about the onset, course, and symptoms of the disorder, and the presence of comorbid diagnoses, with a special focus on mental disorders. The results of diagnostic procedures are collected and if they have not been examined, patients are offered Hcrt-1 examination and HLA typing as a standard part of the diagnostic process.

### 2.1. Epidemiological Data and Prevalence of Psychotic Symptoms and/or Schizophrenia in National Database

Epidemiological data were obtained to determine the prevalence of NT1 and psychotic disorders in Slovakia in 2021. The proportions of NT1 cases with EDS with/without cataplexy and confirmed diagnoses of NT1 ≤ age of 18 years were calculated.

### 2.2. Clinical Picture, PSG, and Other Diagnostic Tests in NT1 Patients with/without Psychotic Symptoms and/or Schizophrenia

In the retrospective analysis of NT1 cases, we identified patients diagnosed with psychotic episode and/or schizophrenia. We evaluated narcolepsy clinical symptoms, results of PSG examination, MSLT, HLA typing, and Hcrt-1 testing in the group of patients with and without psychotic episode.

### 2.3. Case Analysis of NT1 Patients with Psychotic Symptoms and/or Schizophrenia

The cases with a history of diagnosed psychotic episodes were studied, and the patients were invited for reassessment by a somnologist, clinical psychologist, and psychiatrist. The aim of the examinations years after the diagnosis of narcolepsy was to evaluate the persistence of the narcolepsy symptoms and mental disorder. A psychological examination was conducted to assess psychotic symptoms, cognitive abilities, and the personality of patients with narcolepsy. Retrospective analysis partially included the assessment of the administered treatment effect on disease symptoms and its safety.

## 3. Results

### 3.1. Epidemiological Data and Prevalence of Psychotic Symptoms and/or Schizophrenia in a National Database

The prevalence of NT1 in Slovakia in 2021 was 12.7/1 million inhabitants, with 69 recorded cases in the narcolepsy database. The average diagnostic delay was 9.8 (standard deviation 11, median 5, IQR 12) years. We revealed three NT1 patients (4%) in the database diagnosed with psychotic episode. In a single NT1 patient (1.5%), schizophrenia was diagnosed.

### 3.2. Clinical Picture, PSG, and Other Diagnostic Tests in NT1 Patients with/without Psychotic Symptoms and/or Schizophrenia

Three NT1 patients diagnosed with psychotic episode were diagnosed with narcolepsy at age ≤18 years with already present all core symptoms. We could not statistically compare the groups of patients with and without a psychotic episode (3 versus 66) due to the unbalanced number of the patients; Tab 1 attempts to describe the clinical picture and the results of diagnostic examinations in both groups of patients.

Patients without psychotic episode developed EDS with/without cataplexy by age ≤18 years in one third, and only seven (10%) of the NT1 dataset had the disease diagnosed by age ≤18 years. Three (43%) patients diagnosed by the age ≤ of 18 years had dual diagnosis. NT1 diagnosing preceded psychotic manifestation by 1–2 years. NT1 patients with psychosis had a shorter diagnostic delay, fully expressed clinical picture, more frequent nocturnal SOREM, and a number of SOREMs in MSLT at the time of diagnosing. The other variables did not differ at the time of diagnosis in comparison with NT1 patients without psychotic episode and/or schizophrenia (Table 1).

### 3.3. Case Analysis of NT1 Patients with Psychotic Symptoms and/or Schizophrenia

#### 3.3.1. Case 1

Patient M.P., at the age of 18, was referred for the evaluation of daytime sleepiness in 2003 (Epworth Sleepiness Scale, ESS, 18 points). He was describing frequent lapses into sleep at school for the last two years. In addition, he complained of states where he could not move in bed at night. When playing and having fun with his younger cousins, he felt like his knees were broken, or he felt weak for a while. He denied having hallucinations when falling asleep or waking up. In 2005, M.P. underwent PSG examination/night PSG which revealed fragmented night sleep without REM pathology, no present sleep apnea, a normal PLMS index, and the MSLT showed mean Sleep latency 2.5 min with the presence of 4 SOREMs in 4 tests. Genetic examination found present the specific haplotype HLA DQB1*0602. He denied lumbar tap to examine the Hcrt-1 level. Based on typical history supported by PSG findings, he was diagnosed with narcolepsy with cataplexy based on ICSD-2 [12]. In 2004, he was asked to undergo a psychiatric evaluation before starting treatment with modafinil as a standard procedure. The psychiatrist did not catch up with productive symptoms and paranoid or suicidal tendencies, and the treatment with modafinil was prescribed at the dose of 100–200 mg daily. In August 2005, he came for a check-up and complained that he heard voices mocking him and giving orders during the day. Voices bothered him during the day. Difficulties lasted three months. He stopped modafinil in January 2005. Anafranil and risperidone treatment was started. In March 2006, he was hospitalised at a psychiatric clinic for severe behavioural disorders during psychotic experiences.

The clinical psychologist concluded in 2006 that the patient’s personality is affected by psychotic episode, presenting instability in his emotions, and paranoid attitudes, while having no insight into his difficulties. The current intellectual performance belongs to the range of below-average cognitive abilities, including borderline neurocognitive functioning. Until 2022, he was under the care of a psychiatrist, requiring repeated admissions at the Department of Psychiatry. He was treated with antipsychotics. Regular visits to a neurologist confirmed the persistent presence of narcoleptic symptoms. However, stimulant treatment was never reintroduced because of the symptoms of a mental disorder.

In 2022, 19 years after being diagnosed with narcolepsy, he underwent a complex examination by a neurologist, clinical psychologist, and psychiatrist.

In addition to regular visits to a psychiatrist and a neurologist, he was being treated for diabetes (Diabetes mellitus type 2). He tended to abuse alcohol, smoked, and was overweight. He finished elementary school, studied poorly, and then began to attend school but did not finish it due to his mental health complaints. The patient had been on a disability pension his whole life, was self-sufficient, and required the help of another person.

A neurological examination revealed the presence of a severely disturbed sleep–wake cycle. Sleep during the day still had an urgent nature, and the patient fell asleep in inappropriate situations, such as walking. Night sleep was significantly fragmented; during awakenings, he used to smoke and sometimes drink alcohol. He continued to describe rare cataplexy events (knee weakness) triggered by laughter; he had no visual, only auditory, hallucinations and no sleep paralysis.

M.P. presented a fundamentally altered personality by psychosis, including insufficient control of reality, and the psychiatrist concluded that he suffered from schizophrenia. The patient is treated with quetiapine (600 mg/die) and haloperidol (3 mg/die) combined with sertraline (100 mg/die) as an add-on treatment. As a result, the positive symptoms of schizophrenia are under control; however, negative symptoms like dysphoria and alogia became the prominent part of residual symptoms.

#### 3.3.2. Case 2

A paediatric neurologist examined the eleven-year-old patient for fatigue and sleepiness in 2012 (K.S.). She also complained of dizziness, leg cramps, and poor-quality sleep. The paediatrician revealed metabolic syndrome with obesity, hypercholesterolemia, hepatopathy, and beginning puberty. A targeted sleep anamnesis revealed the presence of excessive daytime sleepiness lasting 2 months, without cataplexy, hallucinations, or sleep paralysis. PSG examination showed fragmented nocturnal sleep with low efficiency; without the presence of sleep apnea, the PLMS index was normal. The MSLT test showed a sleep latency of 6 min and the presence of SOREM in 1 of 5 tests. The presence of HLA DQB1*0602 was confirmed. The diagnosis of narcolepsy was not unequivocally confirmed; a re-examination with a latency was recommended. A child clinical psychologist reported increased tension, which was manifested by psychomotor restlessness and mild anxiety. The child psychiatrist assessed the condition as a combination of an exhausting syndrome, a neurotic decompensated child, and sibling rivalry and recommended treatment with SSRI (Fluvoxamine Maleate).

Two years later, she underwent a somnological re-examination. She complained of daytime sleepiness (ESS 16 points), vivid dreams, visual hallucinations while falling asleep, and cataplexy was not present (May 2014). Results of PSG and MSLT with REM abnormalities were evaluated with caution because the patient was taking SSRIs at the time, and a hypocretin level test was recommended. In November 2014, when the lumbar tap was set, she described a strange experience of emotionally triggered muscle weakness and hallucinations (visual, very vivid). She had trouble distinguishing between a dream and reality. Hypocretin-1 in CSF was undetectable (Leiden University), and she was diagnosed with narcolepsy with cataplexy [12]. In December 2014, she was admitted to the psychiatric ward, where police officers brought her from the station. “I talked to God, and he told me to get into a white van; there were Satanists. Some gentleman wanted to catch me, so I kicked him, but they already helped me in the ambulance.” According to the records, she was aggressive, confused, had paranoid delusions, and midazolam was used to calm her down. Because of psychotic symptomatology, she spent 4 weeks in the hospital with an up-titration of aripiprazole to 15 mg daily. The psychiatrist concluded a schizophreniform psychosis. The psychologist found the presence of psychotic episodes due to a pathologically developing personality. A year later (2016), she felt much better; there were no psychotic symptoms present and the psychiatrist decided to reduce the aripiprazole dose and add modafinil up to 200 mg. Psychotherapy was focused on regime organisation.

Up to 2022, the patient was under the careful supervision of a psychiatrist, clinical psychologist, and neurologist. She was still using modafinil and 2.5 mg of aripiprazole. During the springtime, she asked for an unscheduled visit; she described worsening sleepiness. Screening polysomnography excluded sleep-disordered breathing. The psychiatrist and clinical psychologist concluded a transient reactive anxiety and depressive symptoms resulting from a situation at school where she had failed exams and had to quit university studies.

#### 3.3.3. Case 3

M.S. was first referred for examination for EDS at the age of 13. However, according to the parents’ data, he was monitored by a clinical psychologist and psychiatrist from the beginning of his schooling. He had problems with concentration and achievement. The psychologist found an intact intellect, learning disorders, slowed psychomotor pace, dysgraphia, and dyslexia. The psychiatrist concluded the findings as ADHD combined with a residual learning disability, ineffective educational style, and negativistic maladaptive reactions to common burden (2014); the same diagnosis was confirmed later in 2016.

In 2018, he underwent a somnological examination. In the history, apart from irresistible sleep attacks, he described sudden muscle loss triggered by anger lasting 3 min, or transient episodes affecting the upper part of the body and face. Based on PSG, MSLT, and low hypocretin-1 level he was diagnosed as NT1 [2]. The treatment consisted in trying to influence the sleep and wakefulness regime; he used to be under the supervision of a personal assistant at school. During the 2019 holidays, the situation worsened. Cataplexy appeared once a week; he was afraid at night, slept by a lighted lamp, heard voices, and saw spiders and wasps. He complained of tactile hallucinations. He went to bed late. He was tight-lipped and stopped looking people in the eye. He repeatedly assaulted his father, “because he had a small room”. He was admitted to the psychiatry department with the conclusion of a severe depressive episode with psychotic symptoms. Therapeutic trials with different antidepressants failed (during the treatment with vortioxetine, he developed euphoria), and the development of tardive dyskinesias accompanied treatment with olanzapine. His parents decided to stop any medication.

During the scheduled visit in 2022, core symptoms of NT1 were present: daytime sleepiness (ESS 12 points) was relatively compensated by a strict sleep–wake schedule with daily exercise and avoidance of emotional triggers, and typical cataplexy was present once a week. He had a realistic view of vivid dreams and hallucinations. He lost 15 kg. Conflicts with parents were not confirmed by motor aggression. The clinical psychologist concluded in 2022 a disharmonic personality development affected by a significantly below-average cognitive ability. M.S. is attracted by external stimuli, could be easily tempted, and is characterised by a rich imagination. He is stubborn, responsible, carefree, and uncritical in his personality, producing sexual fantasies. The manifestations of dissimulation appeared without paranoid symptoms. In social functioning, he is passive and dependent on the other person. The marginal position in the group is caused by self-blaming, egocentrism, poor sociability, and non-acceptance by others. The psychotic episode was not proven at this time, as assessed by an analysis of psychopathology, psychiatric examination, and psychodiagnostic procedure as well. Taking into account the present symptoms, the age of the patient, and the course of the disease, the psychiatrist concluded the diagnosis as other mixed disorders of conduct and emotions.

## 4. Discussion

By 2021, 69 cases of NT1 were diagnosed in Slovakia. Three patients in the database were simultaneously diagnosed with psychosis or a psychotic illness. Patients with dual diagnosis had manifested EDS and cataplexy at the age ≤18 years; the diagnostic delay was very short compared to the database population. A detailed retrospective analysis of the cases confirmed the clinical symptoms of narcolepsy and cataplexy supported by PSG and MSLT findings and the presence of a specific haplotype in all of them; even hcrt-1 deficiency was detected in two cases. Routinely performed psychological and psychiatric examinations at the time of diagnosis were without the presence of psychotic symptoms. Psychotic symptoms with behavioural disorders were documented within two years of being diagnosed with NT1. The unique study design was to assess the development of psychotic symptoms over time. An examination years later made it possible to evaluate the psychotic episode as solitary in two patients, while schizophrenia was confirmed in one patient. A history of schizophrenia significantly influenced the symptomatic treatment of NT1.

Mood, anxiety disorders, and attention deficit hyperactivity disorder are frequent psychiatric comorbidities of NT1 [13,14,15]. Psychosis and schizophrenia are rare. Hanin et al. [16] reviewed 100 papers from PubMed and analysed 58 NT1 cases. Psychosis and narcolepsy were identified in two forms. Seventeen patients were diagnosed with the psychotic form of narcolepsy. They experienced more severe and vivid apparently REM-related hallucinations or dream/reality confusions being rationalised in delusion-like experiences. Forty-one patients with comorbid schizophrenia spectrum disorder and psychotic symptoms unrelated to sleep had more disorganised symptoms.

Theories to explain the shared pathophysiology between NT1 and schizophrenia included possible immune-mediated pathophysiology or dopaminergic dysfunction [17]. The prevalence of NT1 and schizophrenia has not been systematically studied. The results of older studies suggested an increased prevalence of NT1 in schizophrenia, and conversely. The findings could not be replicated in well-designed studies, including hypocretin and HLA examinations [18]. Two sleep centres in France and Spain found a history of schizophrenia spectrum psychosis in 1.8% of NT1 patients [19]. The national study expressing the prevalence of NT1 and schizophrenia in Slovakia is consistent with the findings of a study of two sleep centres in France and Spain. Both results are within the frequency range of schizophrenia in the general population, suggesting co-occurrence of the disorders rather than pathophysiological overlap. Manifestation of NT1 followed by a psychotic episode has been observed in cases where NT1 manifested in childhood or adolescence [16,19,20], in agreement with the Slovakian database. Their association may be related to higher vulnerability during the ongoing neurodevelopmental period of life and an ongoing sleep–wake disorder.

Although the first symptoms of NT1 appear during puberty and adolescence in many patients, the diagnosis is usually made later. The reason is most often the fact that the clinical picture is not fully expressed at the onset. Zhang et al. [21], based on machine learning results, suggest that the length of the diagnostic delay depends on the correct recognition of cataplexy. Likewise, the diagnostic delay in the Slovak database was very short in cases where the delay between the start of EDS and the occurrence of cataplexy was short. NT1 symptoms in the presented cases preceded the manifestation of psychosis, the onset of which was delayed by 1–2 years. At the time of the psychotic episode, the clinical picture of NT1 was fully developed, including cataplexy and hypnagogic and hypnopompic hallucinations.

Psychotic-like symptoms in narcolepsy, such as hypnagogic and hypnopompic hallucinations, delusion-like vivid dreams, and REM parasomnias in narcolepsy, could present significant diagnostic challenges [8]. On re-examination many years later, symptoms of EDS and cataplexy were still present in all three patients. Persistent unquestionable cataplexy in case 1 confirmed the comorbidity of schizophrenia and NT1, even if he refused the Hcrt-1 examination. In the other two patients, the clinical picture and course of the disease in the long-term follow-up confirmed that these were solitary episodes of psychosis in the course of Hrct-1 deficient narcolepsy. Long-term observation of the clinical course made it possible to conclude whether it was a solitary psychotic episode in the course of NT1 or schizophrenia, which is essential for treatment management.

Although in all three patients, treatment with stimulants (modafinil in two patients, methylphenidate in one patient) was started after the diagnosis of NT1 was made, we do not assume that it caused the onset of psychotic symptoms, as it was discontinued a few weeks before their appearance due to noncompliance or lack of efficacy. Symptomatic NT1 treatment in a schizophrenic patient (Case 1) never reintroduced. Despite the administration of antipsychotic treatment and active psychotherapy, it was impossible to consolidate his sleep–wake cycle and fundamentally affect the symptoms of NT1. In patients with a solitary psychotic episode in the course of NT1, it was possible to reintroduce stimulants with latency. In case 2, modafinil did not show appropriate efficacy, but later on the patient was successfully treated with pitolisant. Parents of the third case, due to numerous failed therapeutic attempts during the psychotic episode, strictly refused further medical treatment. A fixed sleep and wake regime with physical activity significantly influenced the symptoms of EDS and cataplexy.

Unfortunately, for all three patients, we found a critical lack of an experienced multidisciplinary team focused on the treatment of narcolepsy and concurrent psychotic disorder, which would be able to manage the regime and medicament treatment of such a rare and specific patient population in Slovakia.

## 5. Conclusions

NT1 patients diagnosed at age ≤18 years have a higher risk of psychotic disorder, which may be related to higher vulnerability during the ongoing neurodevelopmental period. In our experience, they must be carefully monitored, especially during the first two years after the diagnosis of NT1. Correct recognition of the symptoms of a psychotic episode is essential for therapeutic management and can prevent worsening of mental illness due to symptomatic treatment of NT1, especially with stimulants. Given that NT1 is a rare disease, our conclusions need to be verified in other national databases.

## Figures and Tables

**Table 1 brainsci-13-00043-t001:** Epidemiological data of NT1 cases in Slovakian Narcolepsy Database, and clinical and polysomnographic characteristics of NT1 with/without psychosis.

**NT1 Cases** (*n*, Males, %)	69 (43 Males, 65%)	
Prevalence in 2021/1 million inhabitants	12.7 (95% Confidence interval 9.8–15.6)	
EDS onset ≤18 years *n* (%)	25 (36%)	
EDS + Cataplexy onset ≤ 18 years *n* (%)	20 (29%)	
NT1 diagnosis ≤18 years *n* (%)	7 (10%)	
**NT1 Cases**	**with Psychosis**	**without Psychosis**
*n* (%)	3 (4%)	66 (96%)
Comorbid schizophrenia in NT1 cases *n* (%)	1 (1.5%)	0
Age at EDS onset Median (IQR) years	12 (8)	23 (17)
Age at EDS + Cataplexy onset Median (IQR) years	14 (7)	24.5 (20.5)
Age at NT1 diagnosing Median (IQR) years	15 (7)	32.5 (18.3)
Diagnostic delay Median (IQR) years	3 (1)	5 (12)
Length of follow-up Median (IQR) years	10 (16)	9 (10)
ESS score Median (IQR)	18 (10)	17 (5)
Cataplexy *n* (%)	3 (100%)	66 (100%)
H/H hallucinations *n* (%)	2 (67%)	35 (53%)
Sleep paralysis *n* (%)	2 (67%)	33 (50%)
Disturbed night sleep *n* (%)	2 (67%)	44 (67%)
REM behaviour disorder *n* (%)	0	11 (17%)
Vivid dreams	3 (100%)	30 (45%)
AHI Median (IQR)/h of TST	2.1 (2.8)	3.15 (8.6)
PLMI Median (IQR)/h of TST	11 (15)	6.1 (10.7)
SOREM in night sleep *n* (%)	3 (100%)	30 (45%)
Mean sleep latency in MSLT Median (IQR) min	3.2 (2.5)	2.9 (2.3)
*n* of SOREMs in MSLT Median (IQR)	4 (3)	3 (2)
HLA typing available *n*	3	42
Presence of DQB1*06:02 *n* (%)	3 (100%)	39 (93%)
Hcrt-1 in CSF (available samples, positive findings, %)	2, 2, 100%	5, 5, 100%

NT1-Narcolepsy type 1, H/H hallucinations-Hypnagogic/Hypnopompic hallucinations, AHI-Apnoea-Hypopnoea Index, EDS-Excessive daytime Sleepiness, ESS-Epworth Sleepiness Scale, PLMI-Periodic Leg Movements Index, MSLT-Multiple Sleep Latency test, SOREM-Sleep Onset REM, HLA Human Leukocyte Antigen Complex, IQR-Inter Quartile Range, TST-Total Sleep Time.

## Data Availability

Data are available from the authors upon request.

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
