# Peer review of "Psychotic Episode and Schizophrenia in Slovakian Narcolepsy Database"

_brainsci, 2022, doi:10.3390/brainsci13010043_

Round 1

Reviewer 1 Report

This is a very interesting study investigating the association between narcolepsy and schizophrenia or psychotic episodes. The paper is of interest for the readers and the journal; however, several changes are recommended before considering it for publication.

The authors are describing the association between narcolepsy type 1 and mental disorders. What about narcolepsy 2? It seems important to describe the differences between both of them in the introduction section.

Abstract

1- In some parts of the abstract the authors are separating differents parts, for instance "conclusions"; however, others are not presented by a heading or subheading.

Introduction

1- In a first part of the introduction, the authors are describing the clinical symptoms occurring in narcolepsy. In a second step, they are describing the biological underpinnings (e.g., HLA-DQB*0602). Please, provide more details about this second part. Is there any biomarkers associated with treatment or prognosis?

2- The section about overlapping symptoms should be expanded. Please, provide more detailed description of it. 

3- The main aims of the paper are to examine how many patients with NT1 had a diagnosis of psychotic symptoms or schizophrenia. This section about "objectives" should be further described in a separate section 1.1.

Material and methods

1- Please, describe with more details the Slovak Narcolepsy Database and how were data provided.

2- Please expand the list of variables that were recorded in the retrospective analysis.

3- Describe ethical aspects in the material and methods sections.

Results

1-Table 1 has a lot of information that can be explained in the results section. Because the group with psychosis is only formed by 3 patients and those without psychosis is formed by 66 patients; it seems irrelevant this table. There is no enough statistical power to compare both groups (they are unbalanced).

Conclusions

1. The conclusions section should not be a summarizing of results. Please provide some recommendations for future research.

Author Response

We want to thank you for the detailed review and comments that improved the submitted paper. Please, find below a detailed description of the inclusion of the comments in the manuscript.

Abstract

1- In some parts of the abstract the authors are separating differents parts, for instance "conclusions"; however, others are not presented by a heading or subheading.

It was a transcription error (subtitle Conclusions) and was erased from the abstract.

Introduction

1- In a first part of the introduction, the authors are describing the clinical symptoms occurring in narcolepsy. In a second step, they are describing the biological underpinnings (e.g., HLA-DQB*0602). Please, provide more details about this second part. Is there any biomarkers associated with treatment or prognosis?

Added:

Despite major advances in understanding of the neurobiological basis of NT1, therapy remains nowadays only symptomatic6. If some biomarkers could specifically target for example the treatment of obesity based on distinct metabolic pathways in NT1 patients, it has to be confirmed in the future7. Hcrt/orexin based therapy provides another challenging therapeutic option, especially nonpeptide orexin receptor agonists, that are currently under development7. The role of immune-based therapies administered as early as possible after disease onset could theoretically slow down the destruction of orexin neurons in some selected patients7.

2- The section about overlapping symptoms should be expanded. Please, provide more detailed description of it. 

Added:

Despite the different characteristics in the type of auditory hallucinations and the higher frequency of visual hallucinations in NT18, determining their origin may not be easy. Other REM sleep abnormalities such as vivid dreams, nightmares and REM sleep behaviour disorder observed in 20-70% of NT1 patients9 could be a source of diagnostic challenges as well.

3- The main aims of the paper are to examine how many patients with NT1 had a diagnosis of psychotic symptoms or schizophrenia. This section about "objectives" should be further described in a separate section 1.1.

We organized the sections as:

2. Materials and Methods

The Slovak Narcolepsy Database combines data from three Sleep centres in Slovakia. It includes patients with a confirmed diagnosis of NT1  according to ICSD2/3  2,12. All patients diagnosed with narcolepsy in the sleep labs are referred for the second opinion to the sleep laboratory and centre for rare hypersomnias at University Hospital of L. Pasteur in Kosice. Patients who signed informed consent (at the moment 100%) undergo extended interview focused on the details about the onset, course, and symptoms of the disorder, the presence of comorbid diagnoses, with a special focus on mental disorders. The results of diagnostic procedures are collected, and if they have not been examined, patients are offered Hcrt-1 examination and HLA typing as a standard part of the diagnostic process.

  1. 1. Epidemiological data were obtained to determine the prevalence of NT1 and psychotic disorders in Slovakia in 2021. The proportions of NT1 cases with EDS with/without cataplexy, and confirmed diagnoses of NT1 ≤ age of 18 years, were calculated.
  2. 2. In the retrospective analysis of NT1 cases, we identified patients diagnosed with psychotic episode and/or schizophrenia. We evaluated narcolepsy clinical symptoms, results of PSG examination, MSLT, HLA typing and Hcrt-1 testing in the group of patients with and without psychotic episode. The cases with a history of diagnosed psychotic episodes were studied and the patients were invited for reassessment by a somnologist, clinical psychologist and psychiatrist. The aim of the examinations years after the diagnosing of narcolepsy was to evaluate the persistence of the narcolepsy symptoms and mental disorder. A psychological examination was conducted to assess psychotic symptoms, cognitive abilities, and personality of patients with narcolepsy. Retrospective analysis partially included the assessment of administered treatment effect on disease symptoms and its safety.
  3. 3. The cases with a history of diagnosed psychotic episode…

Results were given subheadings with small changes due to the content:

3.1. Epidemiological data and prevalence of psychotic symptoms and/or schizophrenia in a national database

…added In a single NT1 patient (1,5%) schizophrenia was diagnosed.

3. 2. Clinical picture, PSG, and other diagnostic tests in NT1 patients with/without psychotic symptoms and/or schizophrenia

Three NT1 patients diagnosed with psychotic episode were diagnosed with narcolepsy at age ≤ 18 years with already present all core symptoms. We could not statistically compare the groups of patients with and without a psychotic episode (3 versus 66) due to the unbalanced number of the patients, Tab 1 attempts to describe the clinical picture and the results of diagnostic examinations in both groups of patients.

Patients without psychotic episode developed EDS with/without cataplexy by age ≤18 years in one third and only 7 (10%) of the NT1 dataset had the disease diagnosed by age ≤18 years. Three (43%) patients diagnosed by the age ≤ of 18 years had dual diagnosis. NT1 diagnosing was preceding psychotic manifestation 1-2 years. NT1 patients with psychosis had a shorter diagnostic delay, fully expressed clinical picture, more frequent nocturnal SOREM and a number of SOREMs in MSLT at the time of diagnosing. The other variables did not differ at the time of diagnosing in comparison with NT1 patients without psychotic episode and/or schizophrenia (Table 1).

3.3. Case analysis of NT1 patients with psychotic symptoms and/or schizophrenia

Material and methods

1- Please, describe with more details the Slovak Narcolepsy Database and how were data provided.

As above

2- Please expand the list of variables that were recorded in the retrospective analysis.

Could you specify, please? We did not want to replicate the results of the database, which was published with the results (up to 2017) in 2020: Narcolepsy in Slovakia – Epidemiology, clinical and polysomnographic features, comorbid diagnoses: a case-control study. Sleep Med. 2020;67:15-22. doi:10.1016/J.SLEEP.2019.10.012

3- Describe ethical aspects in the material and methods sections.

Was added as above, details of the ethical approval are included in the last part of the manuscript based on the instructions

Results

1-Table 1 has a lot of information that can be explained in the results section. Because the group with psychosis is only formed by 3 patients and those without psychosis is formed by 66 patients; it seems irrelevant this table. There is no enough statistical power to compare both groups (they are unbalanced).

With the table we provide we were attempting to describe clinical features, we were aware of the limitation of the sample size for statistical analysis (what we emphasised in the results)- as above

Conclusions

  1. The conclusions section should not be a summarizing of results. Please provide some recommendations for future research.

We corrected as bellow:

NT1 patients diagnosed at age ≤ 18 years have a higher risk of psychotic disorder, which may be related to higher vulnerability during the ongoing neurodevelopmental period. In our experience, they must be carefully monitored especially during the first two years after the diagnosis of NT1. Correct recognition of the symptoms of a psychotic episode is essential for therapeutic management and can prevent the worsening of mental illness due to symptomatic treatment of NT1, especially with stimulants. Given that NT1 is a rare disease, our conclusions need to be verified in other national databases.

Reviewer 2 Report

Article titled “Psychotic episode and schizophrenia in Slovakian Narcolepsy  Database” by Feketeová E et al.,

This article is fascinating and very rare case reports of three NT1 cases. Rare case reports in the literature and our clinical experience of NT1 accompanied by a psychotic disorder gave authors the idea to look up their frequency in the Slovak Narcolepsy Database.

The study aimed to identify the prevalence of psychosis in patients with NT1 and to examine affected individuals many years after diagnosis to determine whether psychotic symptoms were a single-episode manifestation of NT1 or were part of well-established schizophrenia.

Narcolepsy patients share many other neurological disorders. Most narcolepsy patients are undertreated for other conditions or become late for diagnosis.

The manuscript is well-written, and each case was fully documented.

The discussion is appropriate.

Minor:

Line 62 – reference number  “82” should change to the correct reference number.

Author Response

We want to thank you for the detailed review and comments that improved the submitted paper.

Line 62 – reference number  “82” should change to the correct reference number.

It was a mistake of the citation manager, which we did not notice, and we corrected it.

Best regards,

Eva Feketeova and Jozef Dragasek 
